# Awareness and Compliance with the Recommendations of Primary and Secondary Prevention of Cancer in Patients with Inflammatory Bowel Disease

**DOI:** 10.3390/jpm13060913

**Published:** 2023-05-30

**Authors:** Edyta Tulewicz-Marti, Beata Stępień-Wrochna, Katarzyna Maciejewska, Michał Łodyga, Katarzyna Karłowicz, Konrad Lewandowski, Grazyna Rydzewska

**Affiliations:** 1Department of Internal Medicine and Gastroenterology with Inflammatory Bowel Disease Unit, National Medical Institute of the Ministry of Interior Affairs and Administration-Warsaw, 02-507 Warsaw, Poland; beata.stepien.md@gmail.com (B.S.-W.); edu_kate@interia.pl (K.M.); karlowiczkatarzyna@gmail.com (K.K.); dr.k.lewandowski@icloud.com (K.L.); grazka3558@yahoo.pl (G.R.); 2Internal Medicine Department, Grochowski Hospital, 04-073 Warsaw, Poland; michal.lodyga@interia.pl; 3Department of Internal Medicine, Faculty of Health Science, Medical University of Warsaw, 02-091 Warszawa, Poland; 4Collegium Medicum, Jana Kochanowski University, 25-317 Kielce, Poland

**Keywords:** IBD, CD, UC, IBDU, oncology awareness, health education, cancer prevention

## Abstract

Introduction: Patients with Inflammatory Bowel Disease (IBD), including Crohn’s disease (CD) and ulcerative colitis (UC), are at high risk of developing malignancies, so prevention and adherence to cancer screening may improve detection. The aim of this study was to assess compliance with medical recommendations, especially primary and secondary prevention of cancer. Methods: This one-center cross-sectional study was carried out between June and December 2021 amongst patients at the Department of Internal Medicine and Gastroenterology, IBD Division, National Medical Institute of Ministry of Interior Affairs and Administrations, or the outpatient clinic. Patients with IBD were asked to complete an anonymous questionnaire, which included 42 questions concerning lifestyle, cancer risk factors, cancer history, and checkups. Statistical methods: The results of the qualitative variables were expressed as frequencies and percentages. We used Fisher’s exact test and the Chi-squared test. A value of *p* < 0.05 was considered significant. Statistical analyses were performed with the SPSS statistical package. Results: A total of 313 patients were enrolled in the study: 145 women and 168 men. In the group, 182 had Crohn’s disease (CD), 120 had ulcerative colitis (UC), and 11 with IBDU (unclassified IBD). Most participants had a disease duration of over 8 years and received biological treatment, corticoids, and/or immunosuppressive therapy. Amongst respondents, 17% (31) of patients with CD and 25.8% (31) with UC were overweight, and 10.5% (19) with CD and 15.8% (19) with UC were obese (*p* = 0.017). We found that 16.3% of all respondents were smokers (79.6% (144) with CD, 90.8% (109) with UC, and 72.7% (8) with IBDU; *p* = 0.053), and 33.9% declared that they consumed alcohol (39.4% (71) with CD, 26.9% (32) with UC, and 18.2% (2) with IBDU; *p* = 0.045). A total of 25.4% of patients were exposed to UV radiation, but only 18.8% used sunblock. In addition, 58.8% (67) of patients with CD and 35.8% (19) with UC receiving immunosuppressants had regular laboratory tests (*p* = 0.02). Furthermore, 41.4% (46) of patients with UC, 27.1% (49) of patients with CD, and 70.0% (7) of patients with IBDU declared not to perform any dermatological control (*p* = 0.013). A total of 77% of patients had abdominal ultrasound. Out of 52.9% of patients for whom colonoscopy was recommended, only 27.3% had it performed (16.9% (30) with CD vs. 43.1% (50) with UC *p* < 0.001). Most examinations were ordered by gastroenterologists. Female patients had regular breast control (CD, 78.6% (66); UC, 91.2% (52); IBDU, 50% (2); *p* = 0.034), and 93.8% (76) had gynecological examinations. Additionally, 80.2% of patients knew about HPV, but most declared not to be vaccinated. A total of 17.9% of patients had urological control, but most had no important pathology detected. Conclusions: According to our study, many patients are still exposed to risk factors, such as obesity, smoking, and low physical activity, that are modifiable. Laboratory tests in patients with immunosuppressive treatment should be performed regularly. Systematic control, especially dermatological checkups, should be recommended. Additionally, not only gastrologists but also other specialists and GPs should remind patients about regular checkups. Primary prevention, such as HPV vaccinations, should be recommended to all patients.

## 1. Introduction

The primary prevention of cancer includes efforts to prevent or avoid precancerous or cancerous conditions from starting; secondary prevention includes the discovery and removal of precancerous conditions and the removal of small cancers before they metastasize [1]. In the 21st century, Inflammatory Bowel Disease (IBD), including Crohn’s disease (CD), ulcerative colitis (UC), and unclassified conditions (IBDU), has become a global disease, with an accelerating increase in incidence in newly industrialized countries [2,3]. Patients with IBD are at risk of developing malignancies not only related to chronic inflammation but also due to several drugs used in treatment (immunosuppressants or biologics), so prevention and adherence to cancer screening are crucial and may improve its detection [4]. There are data showing relationships between modifiable factors, such as alcohol intake and a higher risk of relapse; cigarette smoking and unfavorable outcomes, including corticoid dependence, surgery, and disease progression; and obesity and the risk of relapse, fatigue, and pain [5]. Furthermore, IBD patients are at risk of some solid cancers, including colorectal cancer, cholangiocarcinoma, and GIST; recent data show a trend toward higher risks of developing hematological malignancies and skin malignancies especially related to IBD therapy [6,7,8]. The ECCO guidelines on IBD and Malignancies include the risk of cancers associated with IBD as well as the risk and management of cancers from therapies used for IBD [9,10], but there is still a need for more real-world data on risk and compliance with recommendations. For the above, it is important that patients follow recommendations, such as a healthy lifestyle, dermatological control, and colonoscopy, but there are still some recommendations that are not well-established. Therefore, the aim of this study was to use a survey to assess the risk and patient compliance with medical recommendations, especially the primary and secondary prevention of cancer, and the awareness of medical personnel regarding the risk associated with the development of screening strategies. 

## 2. Materials and Methods

### 2.1. Study Population

This one-center cross-sectional study was carried out in the gastroenterology department of a tertiary referral center (the Department of Internal Medicine and Gastroenterology, IBD Division, National Health Institute of the Ministry of Interior Affairs and Administrations) in Poland. Between June and December 2021, 313 adult patients who were followed at the outpatient clinic or hospitalized were asked to complete an anonymous questionnaire.

### 2.2. Study Design

The survey included 42 questions concerning demographic characteristics, including sex, age, height, and weight (BMI was calculated afterward); the type of the disease; the duration of treatment; current and previous medication; and the frequency and type of control. In the survey, we also included questions about lifestyle, possible cancer risk factors (physical activity, smoking, alcohol consumption, and sun exposure), and cancer history. Moreover, patients with IBD were asked about checkups (dermatological, colonoscopy, abdominal ultrasound, gynecological, and urological control). Most questions were multiple-choice; only one was an open question. The study protocol, including the survey, was approved by the institutional ethical committee, and all participants provided informed consent prior to study enrollment. 

### 2.3. Statistical Methods

The results of the qualitative variables are expressed as frequencies and percentages. We used Fisher’s exact test and the Chi-squared test. A value of *p* < 0.05 was considered significant. Statistical analyses were performed with the SPSS statistical package.

## 3. Results

A total of 313 patients were enrolled in the study: 145 women and 168 men. In the group, 182 had Crohn’s disease (CD), 120 had ulcerative colitis (UC), and 11 had unclassified disease (IBDU). Most patients had a duration of disease of >8 years. Most patients had received mesalazine, steroids, immunosuppressants, and biological treatments; these data are shown in Table 1.

In the group of patients, almost 59.3% of patients had a normal weight, 5.1% were underweight, and, surprisingly, 29.8% were overweight and 12.5% were obese. In the CD group, 61.3% had a normal weight, and in the UC group, 55.0% had a normal weight. In addition, 11% of patients with CD and 3.3% of patients with UC were underweight. Furthermore, 25.4% of patients with CD, 37.5% of patients with UC, and 18.2% of patients with unclassified IBD were overweight. A total of 2.3% of patients with CD, 4.1% of patients with UC, and 9.1% of patients with unclassified IBD were obese (Figure 1). In addition, 122 respondents declared that they had a family history of cancer.

Patients were asked about medications taken in the past (Table 2) and currently (Table 1) and whether they controlled biochemical parameters while taking mesalazine (creatinine and morphology) and immunosuppressive treatment (creatinine, morphology, and transaminases); see Figure 2.

It was observed that 76.7% of patients who took 5-ASA and 50.3% of patients who took immunosuppressive treatments responded and proceeded to regular laboratory control. Additionally, 72.2% of respondents with CD and 86.8% with UC had a controlled peripheral blood count, aminotransferase activity, and kidney function. There was a significant difference between CD and UC in the vigilance of immunosuppressive treatment (*p* = 0.002). 

### 3.1. Physical Activity Status, BMI, Smoking, and Alcohol Intake

In the survey, patients were asked about their physical activity status, smoking, and alcohol intake. A total of 53.7% of patients declared that they did not perform any physical activity: 107 patients with CD, 58 with UC, and 3 with IBDU (Figure 3). 

A total of 83.7% of patients declared that they did not smoke; however, 10.3% of patients responded that they smoked cigarettes; 1% used IQOS, 3.8% used e-cigarettes, and 1.3% used other products with nicotine. In addition, 20.4% of patients with CD, 9.2% with UC, and 27.8 with IBDU declared that they smoked any products containing nicotine. In general, 33.9% of patients declared that they consumed alcohol (39.% of CD patients, 26.9% of UC patients, and 18.2% of IBDU patients). 

In the whole study group, 74.6% of patients declared that they avoided sunlight, whereas 18.8% responded that when exposed to sunlight, they used sunblock; 6.3% did not use sunblock, and one person used a sun bed. Amongst CD patients, 17.0% declared that they used sunblock when sunbathing, and 7.4% did not use sun protection when they suntanned. A total of 22.4% of patients with UC used sunblock, 5.2% did not, and one patient used a sunbed. In the IBDU group, 9.1% used sunblock, and others avoided sunlight (Figure 4).

Amongst all respondents, dermatological control was performed every 6 months in 16.9% of patients, every 12 months in 33.8% of patients, every 2 years in 15.6% of patients, and never in 33.8% of patients (*p* = 0.0013). In total, 17.7% of patients with CD and 17.1% with UC had dermatological control every 6 months; 35.4% of patients with CD, 32.4% with UC, and 20% with IBDU had dermatological control every 12 months; 19.9% of patients with CD, 9% with UC, and 10% with IBDU had dermatological control every 2 years; and 27.1% of patients with CD, 41.4% with UC, and 70% with IBDU did not have dermatological assessments at all.

In terms of colonoscopy, in general, 52.9% of all respondents had to receive a colonoscopy in the previous year, but only 27.3% of them completed the endoscopy. In addition, 77.2% of respondents had an abdominal ultrasound in the past year (76.3% with CD, 79.1% with UC, and 72.7% with IBDU), 14.5% had an abdominal ultrasound in the past 1–2 years (15.8% with CD, 13% with UC, and 9.1% with IBDU), and 8.3% were not sure (7.9% with CD, 7.8% with UC, and 18.2% with IBDU). A total of 10.2% of respondents had an MRI ordered (11.3% with CD and 9.6% with UC; Figure 5).

Most of the medical examinations were ordered by a gastroenterologist; the rest were ordered by a GP or another doctor (Figure 6).

### 3.2. Breast Cancer, HPV-Related Dysplasia, and Cancer of the Uterine Cervix Prevention 

Female patients were asked about breast control, cytology, gynecological ultrasound, and HPV vaccination. A total of 91.2% of UC patients and 78.6% of CD patients declared that they had their breasts examined (*p* = 0.034); breast ultrasound was performed in 59.6% of UC and 48.8% of CD patients (*p* = 0.4) in the past 12 months (Figure 7). 

A total of 98.8% of respondents declared that they received regular gynecological examinations; 52% of patients with CD and 58.3% with UC declared that they had gynecological cytology in the past year, 26% of patients with CD and 33.3% with UC declared that they had cytology in the past 1–2 years, and 18% of patients with CD and 5.6% with UC had cytology in the past 5 years. A total of 93.6% of patients with CD, 98.2% with UC, and all female patients with IBDU had ever had a gynecological ultrasound performed, and 56.1% of patients with CD, 64.3% with UC, and 75% with IBDU had one in the past 12 months. 

Furthermore, 80.2% of patients with CD, 78.9% with UC, and 75% with IBDU knew about HPV vaccinations, but only 15.2% of patients with CD, 28.6% with UC, and 13.3% with IBDU were vaccinated (Figure 8).

All male patients were asked about urological control, and only 17.9% (14) with CD and 10.9% (6) with UC had urological control (Figure 9).

## 4. Discussion

Our results showed that although IBD patients attached great importance to the control of their health status, they did not always follow our recommendations, such as dermatological control, colonoscopy surveillance, and gynecological control in women.

Surprisingly, in the whole study group, there were overweight and obese patients; some respondents declared that they smoked and consumed alcohol. Taking a deeper look at the responses, an important observation is that only 61.3% of patients had normal BMI, and 11% of patients were underweight; most of them had CD, which is possibly related to malabsorption. According to our data, less than half of the respondents had a BMI higher than normal, and most of them had UC. It has been previously observed that approximately 15–40% of patients with IBD are obese and 20–40% are overweight, which apparently may contribute to the development of IBD, obesity-related gut microbiota, and IBD-related CRC [11,12]. It is important to know that obesity is associated with a lower prevalence of clinical remission and higher anxiety and pain, and exercise may reduce tumor necrosis factor-α and possibly augment the response to antagonists TNF-α [13]. Furthermore, disease activity was associated with obesity in newly diagnosed pediatric patients with UC [14]. On the other hand, in the general population, obesity may be related to different cancers, such as colon, melanoma, multiple myeloma, and leukemia, as well as cancers of the endometrium and breast in women [15]. For this reason, physical activity, BMI control, and cancer screening are extremely important in this group.

Moreover, almost 17% of patients smoked (cigarettes, IQOS, e-cigarettes, and others), and most were patients with CD. It is known that smoking has been associated with more severe and perianal disease and the stricturing phenotype of CD, and it should be prohibited in this group of patients [16]. With all the above in mind, smoking was also shown to be associated with CIN+; therefore, it should be strongly restricted [17]. 

Another observation was that over one-third of patients declared that they consumed alcohol. Alcohol has not been definitely linked with clinical activity; moreover, some studies have demonstrated a protective effect of alcohol consumption with respect to UC development. This effect is negated when drinking alcohol is combined with cigarette smoking [18]. In general, the alcohol-drinking population may have an increased risk of colorectal, breast, and liver cancer [19]. Considering all the above, lifestyle modifications, including eating a healthy diet, maintaining BMI, engaging in physical activity, quitting smoking, and reducing alcohol intake, may improve primary prevention and reduce the risk of cancer, especially in this group of IBD patients.

According to the risk of malignancies related to medications, immunosuppressive treatments, and biologics should be analyzed. According to the CESAME study, patients with IBD with a history of cancer are at an increased risk of developing any (new or recurrent) cancer, with a predominant incidence of new cancers [20]. Keeping that in mind, we should be more vigilant, especially for this group of patients. First, thiopurines are commonly used in IBD. Their mechanism consists of integration into the DNA of proliferating cells, and they promote the inflammatory process. It was observed that thiopurines may increase the risk of hematological disease (especially in older patients), non-melanoma skin cancer (NMSC), cervical cancer, and cancer of the urinary tract [9]. In our cohort, most patients were treated with thiopurines in the past, and less than half were taking thiopurines currently, but only half of the respondents declared that they received checkups for biochemicals and morphology. There are contradictory data concerning the use of thiopurines and the overall risk of cancer [21,22,23]. Even though the use of biological treatment has been rising (70.9% in the whole study group), the potential overall risk of malignancies related to this therapy (especially to TNF-α antagonists) seems to be low, even in older patients [24]. 

It is not clear whether IBD is an independent factor for melanoma, but it was observed that the risk of non-melanoma skin cancers (NMSC) has risen [10]. Moreover, advanced age is associated with NMSC. According to our study, one-fourth may be at risk of skin lesions. Most patients declared that they did not sunbathe regularly; however, some still declared that they sunbathed with sunblock protection (more often UC than CD patients), and a few sunbathed without. One person used a sunbed. According to the ECCO recommendations, IBD patients, especially those who are on thiopurine treatment, have a higher risk of NMSC and should undergo skin-cancer screening and take sun-protective measures [9,10]. In our cohort, 33.8% of patients underwent annual skin screening (35.4% with CD vs. 32.4% with UC), and 16.9% of patients underwent screening every 6 months. Still, over one-third of patients had not undergone skin cancer screening recently. Moreover, in our cohort, gastroenterologists mainly recommended dermatological screening, but also it was recommended by dermatologists and other doctors. Still, almost one-fifth of patients requested skin cancer screening themselves.

Regarding the surveillance of cholangiocarcinoma, PSC, and solid tumors, most patients had abdominal ultrasounds, which is a repetitive technique and enables the assessment of the structures of the abdomen. Most respondents had an abdominal ultrasound recently, in the past year. It was mostly recommended by gastroenterologists and less often by other doctors of other specialties. Only 10% of patients had MRI recommended, and it was mostly recommended by gastroenterologists; however, this test is reserved for accurate diagnosis, especially of diseases of the bile ducts, and it is not used on a large scale. Over half of respondents had a colonoscopy recommended; of them, only 27% had it performed. It was mostly recommended by gastroenterologists and GPs. Knowing the fact that patients with IBD have an increased risk of colon cancer, surveillance colonoscopy in patients with IBD is important; according to the ECCO guidelines, the first colonoscopy should be on 8 years after the onset of symptoms, and the next one, depending on the risk factors, should be performed 1–5 years later [9]. To our knowledge, our study is one of the few to evaluate adherence to endoscopic probes. According to most studies, adherence to colonoscopy is less than 30% [24,25,26], which is similar to our study results.

Apparently, most female patients received gynecological check-ups, and half of them declared that they had cytology performed in the past year. According to recent data, it remains unclear whether immunosuppressants are associated with a greater risk of cervical dysplasia in IBD than not receiving immunosuppression; however, all female patients should be encouraged to receive cervical cancer screening [9,27]. Respondents claimed to perform a self-check of the breasts (however, patients with UC performed self-checks statistically more often than those with CD) and breast ultrasound, and some of them also had mammography. Most respondents knew about HPV but were not vaccinated. According to recent data, female patients who are current smokers, have an age at diagnosis of <20 years, have extensive disease, and have been exposed to >10 prescriptions of oral contraceptives are at risk for HPV-related cancer and dysplasia of the uterine cervix. Moreover, preventive measures, such as Pap test screening and HPV vaccinations, should be performed in this group of patients [9]. 

Less than 20% of patients declared that they received urological control, and only 11% had some alterations detected. There are little data on the risk of prostate cancer in IBD patients; however, according to a systematic review by Haddad et al., men with UC appear to have a higher risk of developing prostate cancer. In patients with CD, this is inconclusive [28,29,30]. Even though it is not strictly recommended to follow urological controls, male patients, especially those with UC over 50 years old, should be considered to be controlled by urologists. 

The present study has several limitations. One is the anonymous survey and data, which were based on the patients’ answers and not on objective data. Second, the survey was mostly completed in the hospital, and many patients were receiving corticoids so they might have had an exacerbation of the disease, which might have influenced the responses. The third is the cross-sectional nature of this study and the lack of follow-up. Further studies should be designed to assess awareness and compliance with the presented recommendations.

## 5. Conclusions

Many patients are still exposed to modifiable risk factors, in particular obesity, smoking, alcohol, and low physical activity. Regular checkups, such as dermatological, gynecological or urological visits, abdominal ultrasounds, and endoscopies, should be ordered, not only by gastrologists but also by other specialists. Primary prevention, such as HPV vaccinations, should be recommended to all patients.

## Figures and Tables

**Figure 1 jpm-13-00913-f001:**
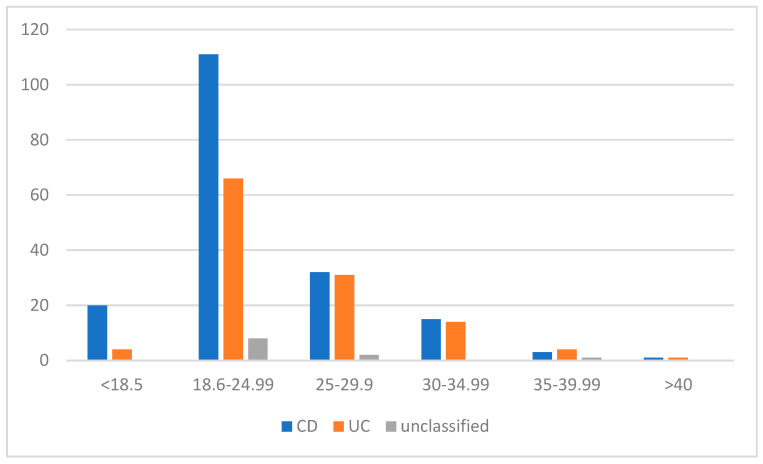
BMI characteristics in IBD patients. Abbreviations: *n*: number of patients, CD: Crohn’s disease, UC: ulcerative colitis, BMI: body mass index (<18.5: underweight; 18–24.99: normal range; 25–29.99: overweight; 30–34.99: obese class I; 35–39.99: obese class II; >40: obese class III).

**Figure 2 jpm-13-00913-f002:**
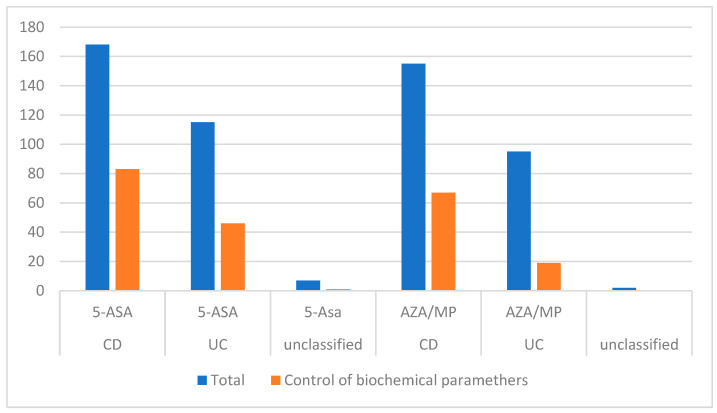
Monitoring of biochemical parameters in patients receiving 5-ASA and immunosuppressive treatment. Abbreviations: CD: Crohn’s disease, UC: ulcerative colitis, 5-ASA: 5-aminosalicylates, AZA/MP: azathioprine/mercaptopurine.

**Figure 3 jpm-13-00913-f003:**
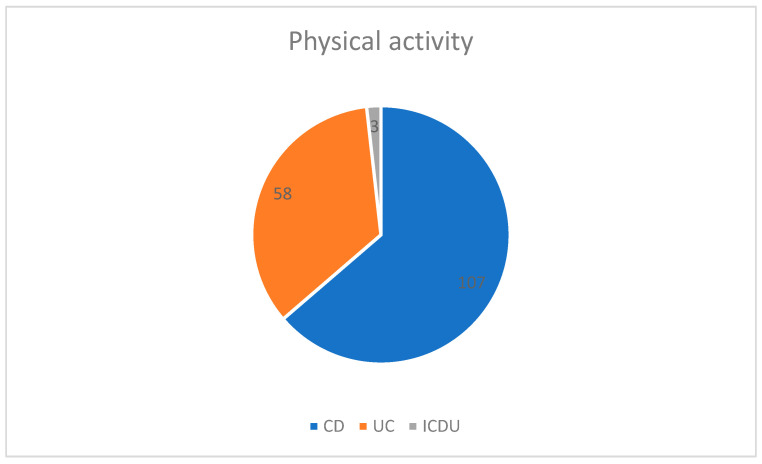
Physical activity in patients with IBD. Abbreviations: CD: Crohn’s disease, UC: ulcerative colitis, IBDU: unclassified IBD.

**Figure 4 jpm-13-00913-f004:**
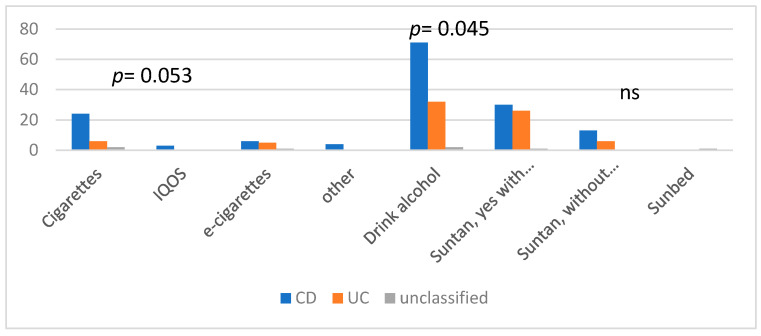
Smoking, alcohol, and sunbathing in IBD patients. Abbreviations: CD: Crohn’s disease, UC: ulcerative colitis, IQOS: “I quit ordinary smoking” product, e-cigarettes: electronic cigarettes.

**Figure 5 jpm-13-00913-f005:**
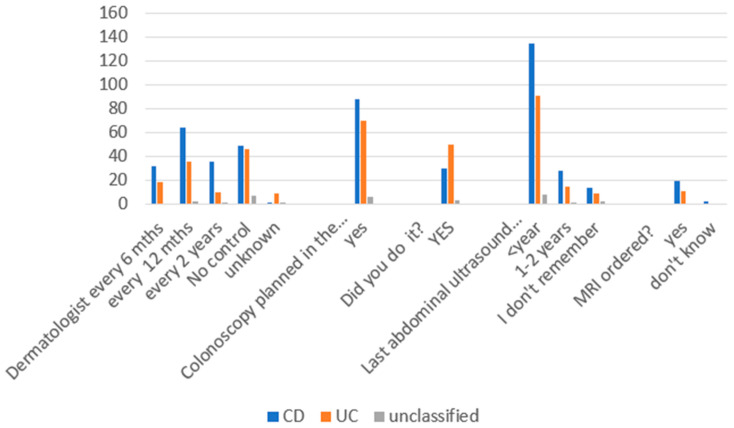
Dermatological control, endoscopy surveillance, abdominal ultrasound, and MRI.

**Figure 6 jpm-13-00913-f006:**
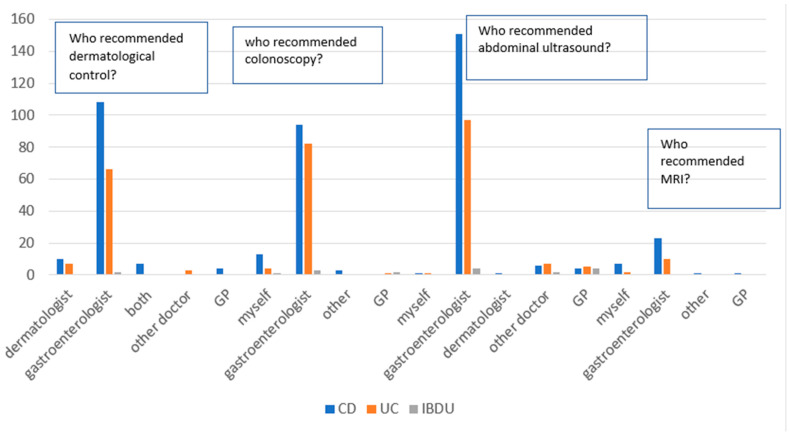
Dermatological control, endoscopy surveillance, abdominal ultrasound, and MRI by specialists who ordered medical examinations. Abbreviations: CD: Crohn’s disease, UC: ulcerative colitis, IBD: IBD unclassified, GP: general practitioner.

**Figure 7 jpm-13-00913-f007:**
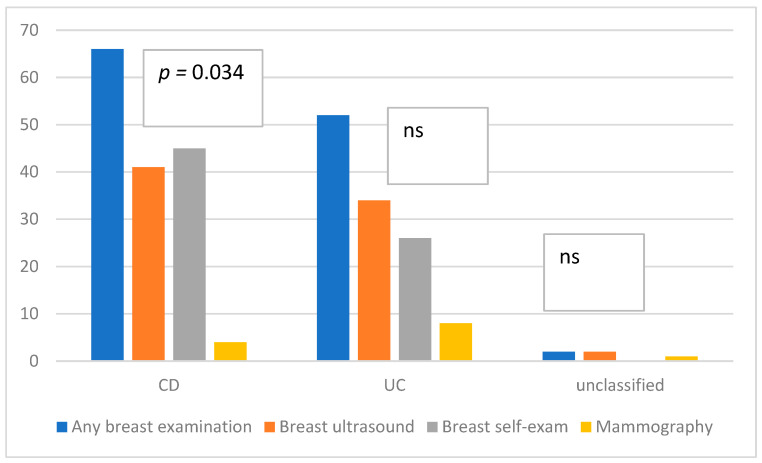
Breast control in female patients. Abbreviations: CD: Crohn’s disease, UC: ulcerative colitis.

**Figure 8 jpm-13-00913-f008:**
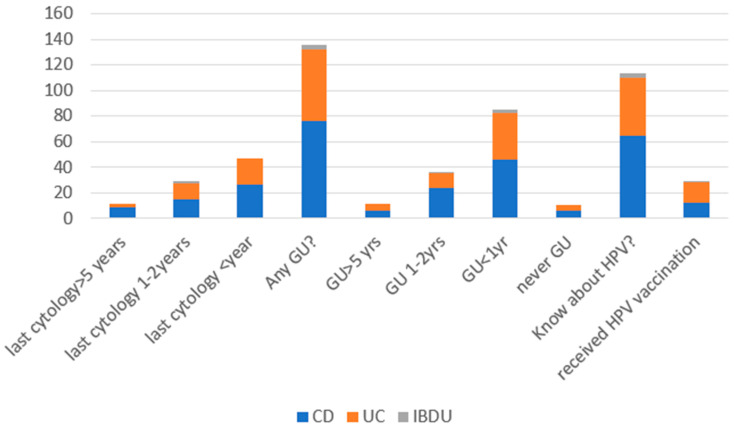
Gynecological cytology, gynecological ultrasound, knowledge about HPV vaccination, and urological control in female patients. Abbreviations: CD: Crohn’s disease, UC: ulcerative colitis; GU: gynecological ultrasound; HPV: human papillomavirus.

**Figure 9 jpm-13-00913-f009:**
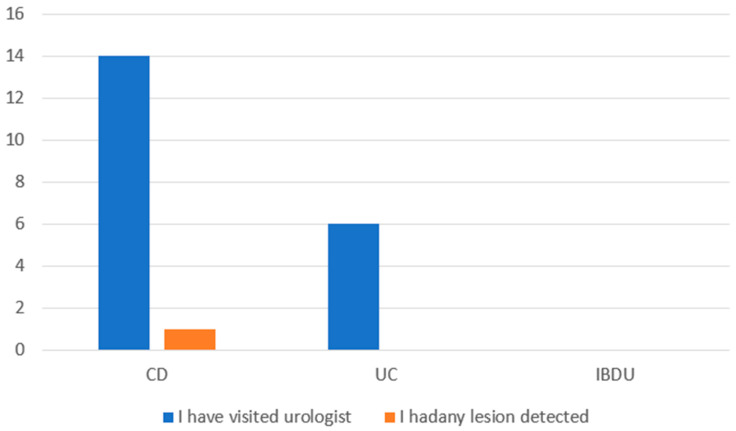
Urological control in male patients. Abbreviations: CD: Crohn’s disease, UC: ulcerative colitis; IBDU: IBD unclassified.

**Table 1 jpm-13-00913-t001:** Patient demographics at baseline.

	CD *n* = 182	UC *n* = 120	IBDU *n* = 11
Female, *n* (%) Male, *n* (%)	84 (46.2%) 98 (53.8%)	57 (47.5%) 63 (52.5%)	4 (36.4%) 7 (63.6%)
Age (years) (%)			
<20	10 (5.5%)	2 (1.7%)	0 (0%)
21–30	63 (34.1%)	35 (29.2%)	1 (9.1%)
31–40	55 (30.2%)	36 (30.0%)	3 (27.3%)
41–50	39 (21.4%)	21 (17.5%)	3 (27.3%)
51–60	11 (6.0%)	17 (14.2%)	3 (27.3%)
61–65	2 (1.1%)	0 (0%)	0 (0%)
>65	3 (1.6%)	9 (7.5%)	1 (9.1%)
Duration of disease (years)			
<1 year	9 (4.9%)	5 (4.2%)	2 (18.2%)
1–5	19 (10.4%)	35 (29.2%)	3 (27.3%)
5–8	39 (21.4%)	27 (22.5%)	2 (18.2%)
>8	115 (63.2%)	53 (44.2%)	4 (36.4%)
Current use of medications			
5-ASA (Mesalazine or sulfasalazine)	168 (92.3%)	115 (95.8%)	7(63.6%)
Corticosteroids			
Prednisone or methylprednisolone	102 (56.0%)	72 (60.0%)	1 (9.1%)
Budesonide	86 (47.3%)	67 (55.8%)	2 (18.2%)
Immunosuppressants			
Azathioprine/6-mercaptopurine	155 (85.2%)	95 (79.2%)	2 (18.2%)
Methotrexate	34 (18.7%)	3 (2.5%)	1 (9.1%)
Biologic agent	149 (81.9%)	71 (59.2%)	2 (18.2%)
Contraceptive/hormone replacement therapy	28 (15.4%)	12 (10.0%)	0 (0.0%)
Other medication	2 (1.1%)	5 (4.2%)	0 (0.0%)

Abbreviations: *n*: number of patients, CD: Crohn’s disease, UC: ulcerative colitis, IBDU: unclassified; 5-ASA: 5-aminosalicylates.

**Table 2 jpm-13-00913-t002:** Treatment received in the past.

Characteristic	Overall, *n* = 313 ^1^	CD, *n* = 182 ^1^	UC, *n* = 120 ^1^	IBDU, *n* = 11 ^1^
Mesalazine, sulfasalazine	290 (92.7%)	168 (92.3%)	115 (95.8%)	7 (63.6%)
Azathioprine, 6-MP	252 (80.5%)	155 (85.2%)	95 (79.2%)	2 (18.2%)
MTX	38 (12.1%)	34 (18.7%)	3 (2.5%)	1 (9.1%)
Budesonide	205 (65.5%)	104 (57.1%)	95 (79.2%)	5 (4.5%)
Prednisone	175 (55.9%)	102 (56.0%)	72 (60.0%)	1 (9.1%)
Biological treatment	222 (70.9%)	149 (81.9%)	71 (59.2%)	2 (18.2%)
Contraceptive/ hormone replacement therapy	40 (12.8%)	28 (15.4%)	12 (10.0%)	0 (0.0%)
Other medication	7 (2.2%)	2 (1.1%)	5 (4.2%)	0 (0.0%)

Abbreviations: *n*: number of patients, CD: Crohn’s disease, UC: ulcerative colitis, IBDU: inflammatory bowel disease, unclassified; 6-MP: 6-mercaptopurine; MTX: methotrexate. ^1^ n (%).

## Data Availability

Data supporting reported results are available upon request.

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
