# Peer review of "Awareness and Compliance with the Recommendations of Primary and Secondary Prevention of Cancer in Patients with Inflammatory Bowel Disease"

_jpm, 2023, doi:10.3390/jpm13060913_

Round 1
Reviewer 1 Report
In this study, the Authors the aimed to assess the risk and patients compliance with medical recommendations, especially primary and secondary prevention of cancer and the awareness of medical personnel regarding the risk associated with the development of screening strategies.
The authors requested anonymous responses from patients on a questionnaire about lifestyle, cancer risk factors, past cancer experiences, and checks in IBD patients.
Where this study was conducted—in an outpatient or inpatient setting, or both. The study population had a high rate of steroid and budesonide use, making the group of patients who were being interviewed appear to be hospitalized; the authors may have described this.
In terms of colonoscopy in general 52.9% of all respondents did have to do colonoscopy in the previous year, but only 27.3% of them did the endoscopy.
Nearly one-fourth of respondents who received a recommendation for a colonoscopy also had one performed. This is a significant finding that the authors should have emphasized as well as the need for more research on the subject.
53.7% of patients declared to do any physical activity. In the whole studied group, 74.6% declared to avoid sunlight. 77.2% of respondents had abdominal ultrasound in the last year. 10.2% of respondents had MRI ordered, 11.3% with CD, 9.6% with UC. 91.2% of UC patients and 78.6% of CD declared to have breasts examined, breast ultrasound did 59.6% of UC and 48.8% of CD in the last 12 months. 98.8% of respondents declared to perform regular gynecological examinations. 80.2% of patients with CD, 78.9% with UC and 75% with IBDU knew about HPV vaccinations but only 15.2% with CD, 28.6% and 13.3% with IBDU were vaccine. 17.9% (14) with CD and 10.9% (6) UC had urological control.
The discussion again summarizes the presented findings (above mentioned), but there is almost no dispute or debate. The authors should compare their reported findings to those in the literature. If there is no any paper or specific topic, the author should additionally state that.
Author Response
Dear Editor,
Thank you for the comments on the article here are the following answers to the revision:
I was asked where this study was conducted—in an outpatient or inpatient setting, or both.
Thank you for this comment, I have corrected that both were asked to participate in the study. Also basing the survey mostly in the hospital many patients were receiving steroids and I have put in the limitations of the study
In terms of colonoscopy in general 52.9% of all respondents did have to do colonoscopy in the previous year, but only 27.3% of them did the endoscopy. I have added some comments on this topic and discussed. I also have compared mentioned findings with the literature
Thank you
Reviewer 2 Report
Interesting study attempting to shed light into a difficult and controversial issue, that of cancer prevention in IBD patients. Although the authors should be commented for their effort there are some issues that need to be addressed.
1) The Discussion is too long, should be re-written and should be re-structured. The authors repeat in the Discussion their findings, although these are already presented in the Results. The Discussion should focus only on the important findings and analyze what previous reports have shown on this matter. Please do not repeat all the results in the Discussion section.
2) The issue of occurrence of malignancy in patients receiving immunomodulators or biologics is far too important and far too serious to be addressed in such a small study. I would it find it more appropriate that authors modify the Discussion accordingly accordingly and concentrate on the predisposing factors studied (for example I found completely irrelevant the comment regarding the Epstein Barr status of the patients receiving thiopurines - i.e. the comment is correct but it has nothing to do with this study).
3) In general the Discussion should be more focused in order for the readers to understand what new this study has to offer.
No comments
Author Response
Dear Editor,
Thank you for the revision here are comments on it
I have rewritten the discussion and also tried to focus on the important findings as it was suggested
“The issue of occurrence of malignancy in patients receiving immunomodulators or biologics is far too important and far too serious to be addressed in such a small study. I would it find it more appropriate that authors modify the Discussion accordingly and concentrate on the predisposing factors studied (for example I found completely irrelevant the comment regarding the Epstein Barr status of the patients receiving thiopurines - i.e. the comment is correct but it has nothing to do with this study). “According this comment I have modified the discussion it and cancelled the EBV comment knowing the fact that it wasn’t the aim of the survey to evaluate this issue
Thank you
Round 2
Reviewer 2 Report
No more comments
No comments